# Post-Harvest Quality and Sensory Evaluation of Mini Sweet Peppers

**Renata Mussoi Giacomin** [1], **Leonel Vinícius Constantino** [1,2], **Alison Fernando Nogueira** [1],
**Maria Beatriz Cadato Ruzza** [2], **Ariele Maria Morelli** [2], **Kelvin Shinohata Branco** [1], **Lais Martins Rossetto** [1],
**Douglas Mariani Zeffa** [3] **and Leandro Simões Azeredo Gonçalves** [1,*]

1   Departamento de Agronomia, Universidade Estadual de Londrina, Rodovia Celso Garcia Cid, PR 445,
    Km 380, Londrina 86051-900, Brazil; giacomin.rm@gmail.com (R.M.G.);
    leonelconstantino28@gmail.com (L.V.C.); alllisonfernando@gmail.com (A.F.N.);
    kelvinsbranco@gmail.com (K.S.B.); laisrossetto@ymail.com (L.M.R.)
2   Departamento de Química, Universidade Estadual de Londrina, Rodovia Celso Garcia Cid, PR 445, Km 380,
    Londrina 86051-900, Brazil; maria.beatriz@uel.br (M.B.C.R.); ariele@uel.br (A.M.M.)
3   Departamento de Agronomia, Universidade Estadual de Maringá, Av. Colombo, 5790,
    Maringá 87020-270, Brazil; douglas.mz@hotmail.com
*   Correspondence: leandrosag@uel.br

**Abstract:** Sweet pepper (*Capsicum annuum* L.) is one of the most consumed vegetables in the world, being recognized as a food with high nutritional value. Recently, the market for sweet and colorful mini peppers has increased, especially among the most demanding consumers in the novelties in vegetables and functional foods. In this sense, we evaluated mini sweet peppers genotypes (Akamu, Kaiki, Kalani, Kaolin e Moke from Isla® seeds) regarding the physical-chemical, nutritional and sensory analysis aspects. A wide variability was observed among genotypes, highlighting the Kalani genotype for total carotenoids, and the genotypes Akamu, Kaiki and Kaolin for phenolic totals content and antioxidant activity. Moke and Kaolin showed higher vitamin C content and fruit firmness. Based on sensory analysis, Kalani, Kaiki, Kaolin and Akamu obtained greater global acceptance. The genotypes can be considered an important marketing strategy of mini sweet peppers trade, associating different shapes, colors and nutritional quality.

**Keywords:** *Capsicum annuum* L.; functional food; pepper pre-breeding; horticulture; sensory analysis

## 1. Introduction

According to the World Health Organization (WHO), non-communicable diseases (NTDs) are the leading causes of death in the world, with ~80% of these deaths occurring in low- and middle-income countries. At least, one third occur among individuals under 60 years, while in high-income countries this proportion is 13% [1]. The main risk factors associated with NTDs include inadequate diets, low nutrition, and lack of physical activity. On the nutritional issue, low consumption of fruits and vegetables is considered an important risk factor that has coantributed to the increase in the overall burden of chronic diseases [2]. It is estimated that up to 2.7 million lives could be saved each year if fruit and vegetable consumption were increased [3].

In Brazil, 55.7% of adults (18 years) are overweight. These, 19.8% have obesity associated an inadequate eating habits, such as high consumption of ultra-processed foods and low consumption of minimally processed and fresh foods [4]. Several factors are related to this inadequate eating habit, including economic barriers, lack of nutritional knowledge and awareness, dietary preferences, and cultural factors [5,6]. In this context, several public policies and innovations in the agricultural sector have been carried out in order to encourage the consumption of vegetables and fruits in the world [2,7].

Mini and baby vegetables are considered one of the alternatives to increase the healthy foods consumption, mainly for children and young people, because they provide greater

ease for consumption and many of them are tastier, making it more attractive both visually and by taste [8]. The horticultural sector is constantly subjected to a process of change that requires the implementation of strategies for innovation of its products. An alternative to diversify this production chain is the cultivation of mini-vegetables which has presented advantageous economic opportunities [7,9]. The popularization of mini-vegetables began in the 1990s in Europe and expanded throughout the world [8].

Mini vegetables can be obtained through genetic improvement or are reduced by processing, while baby vegetables are obtained by pre-harvesting the traditional-sized product [9]. The best known crops of this segment in the Brazilian market are the mini tomatoes (*Solanum lycopersicum* L.), mini carrots (*Daucus carota* L.), mini lettuce (*Lactuva sativa* L.) and baby leaf, which includes lettuce, watercress (*Nasturtium officinale* R.), beetroot (*Beta vulgaris* L.), arugula (*Eruca sativa* L.), among other species [9].

Sweet pepper (*Capsicum annuum* L.) is among the ten most consumed vegetables in Brazil and in the world [10,11]. In addition to the attractions such as color and aroma, this vegetable is well known for its chemical and nutritional properties. Its fruits are great sources of vitamins A, B, C and E, as well to bioactive compounds with antioxidant activity like carotenoids and phenolic compounds [12,13]. Recently, mini sweet peppers of different colors, sizes and shapes have been introduced in the Brazilian market. However, there is lack of information about its physical-chemical and nutritional properties. Thus, the present work aimed to evaluate mini sweet peppers genotypes by physical-chemical, nutritional and sensory aspects.

## 2. Materials and Methods

### 2.1. Plant Material and Experimental Design

The experiment was conducted at the Agronomy Department of the Universidade Estadual de Londrina (UEL), Londrina, Paraná, Brazil. Five genotypes of mini peppers (Akamu, Kaiki, Kalani, Kaolin and Moke) from the company Isla Sementes (Figure 1) were evaluated, which were sown in polystyrene trays, on Subras® commercial substrate. After 30 days, the seedlings were transplanted to pots, containing a mixture of soil and sand in a 3:1 ratio. The experimental design was complete randomized blocks with 3 replications and 12 plants per plot. The plants were kept in protected cultivation following the practices recommended for the cultivation of sweet peppers in Brazil. The fruits were harvested ripe (55 days after anthesis) and submitted to physical characterization. They were then stored under refrigeration at 4–6 °C for up to 3 days before biochemical analyses.

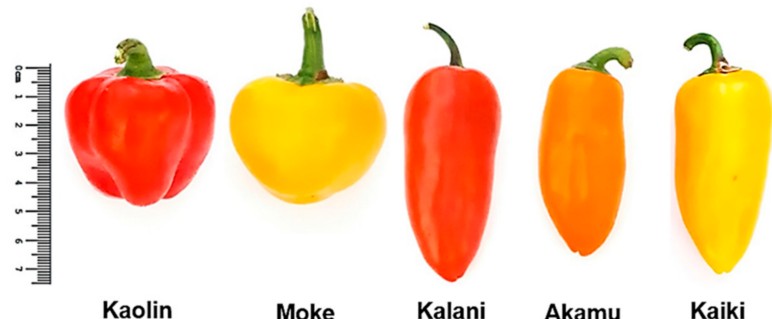

**Figure 1.** Photograph of the five mini sweet peppers evaluated in the present study.

### 2.2. Physical Characterization

Eight ripe fruit (55 days after anthesis) of each genotype were harvested and evaluated according to descriptors established by IPGRI [14], currently Bioversity International: length, diameter, pericarp thickness, mass and dry mass content. The color of the fruits was determined in a colorimeter using illuminant D65 (Minolta Co., Tokyo, Japan, model CR-13) by the luminosity, Chroma and Hue angle.

The firmness of five fruits of each genotype was determined in newtons (N) by the puncture test in a texturometer (Model TA. XT Plus, Stable Micro System, Surrey, UK). A needle probe was used to measure the resistance of the exocarp (skin) and pericarp. The puncture speed was 0.5 mm s$^{-1}$ until 5 mm of the fruit was perforated in the equatorial zone of each fruit with no removed peel.

### 2.3. Biochemical Characterization

Soluble solids content (SS) was determined in a portable digital refractometer (Atago$^{®}$) and expressed in °Brix. Titratable acidity (TA) was quantified by titrimetry based on AOAC method 942.15 [15] and expressed as% citric acid. Also was calculated the SS/TA ratio.

Vitamin C content was quantified by the titration method based on [15] and modified by [16], expressed as mg ascorbic acid 100 g$^{-1}$. Extraction of total carotenoids was adapted from Adalid, Roselló and Nuez [17] and quantification performed according to [18] using in a spectrophotometer (Genesys 10, Thermo, Waltham, MA, USA) at 450 nm, reporting as µg beta-carotene equivalents 100 g$^{-1}$.

For the quantification of phenolic compounds and antioxidant activity, an extraction was made from 1.0 g of fresh samples with 10 mL of 70% (*v/v*) ethanol, leaving the suspension under stirring for 30 min (Orbital-Nova Orgânica) at room temperature (25 °C). Then, the extract was centrifuged at $1013 \times g$ (Excelsa 2 Fanem model 205N) for 5 min and separated for analysis [18].

The extraction of total phenolic content, total flavonoid content and antioxidant activity was performed according to [18]. The quantification of phenolic compounds was based to [19], where Gallic acid was used as standard ranging from 10 to 100 mg L$^{-1}$ (r = 0.9960) and expressed as mg gallic acid equivalents (GAE) per 100 g of fresh mass.

The antioxidant activity by sequestration of the 2,2-Diphenyl-1-picryl-hydrazyl (DPPH·) radical was performed according to Brand-Williams, Cuvelier and Berset [14]. Trolox (6-hydroxy-2,5,7,8-tetramethylchroman-2-carboxylic acid) was used as standard ranging from 0.20 to 1.00 mmol L$^{-1}$ (r = 0.9992), and the result was expressed in % free radical scavenging. For the FRAP assay, 0.05 mL of the ethanolic extract were used, mixed with 1.0 mL of 80% (*v/v*) methanol and 1.0 mL of the FRAP reagent. The mixture was maintained at 37 °C for 30 min and the reading was performed at 595 nm, expressed as µmol TEAC per 100 g [20]. The ABTS test consisted of a mixture of 2 mL of the diluted ABTS•+ solution with 50 µL kept at darkroom temperature at 35 °C for 5 min. The absorbance was read at 753 nm. Trolox was used as standard and the result expressed as µmol of TEAC per 100 g [21].

### 2.4. Sensory Evaluation

The acceptance test was applied in a single session to a group of 155 evaluators (untrained volunteers), members of the university community of the Universidade Estadual de Londrina, Londrina, Paraná, Brazil. The proposal was approved by the Committee for Ethics in Research on Human Beings under registration CAAE 98049918.8.0000.5231.

The evaluators received one sample at a time, encoded with three random digits, served on a transparent disposable plate containing one fruit cut. The samples evaluation was done using a 10 cm hybrid hedonic scale anchored in the middle and extreme regions of the scale (0 = disliked extremely, 5 = neither liked nor disliked, 10 = liked extremely) (Villanueva et al., 2005) for the attributes: size, shape, color and overall acceptance. To verify the purchase intention of the product, a 5-point structured scale was used (1 = certainly would not buy, to 5 = certainly would buy).

### 2.5. Data Analysis

The data were submitted to the F-test ($p \leq 0.05$) by analysis of variance (ANOVA) and means were compared using the Tukey test ($p \leq 0.05$). Principal component analyzes (PCA) and Ward's hierarchical grouping were also performed. All statistical analyzes were performed using software R version 3.6.0 [22] using packages ExpDes [15] FactoMiner [16], pheatmap [17] and e ggplot2 [18].

## 3. Results

### 3.1. Physical and Biochemical Characterization

Analysis of variance showed a significant effect ($p \leq 0.05$) for all evaluated characteristics, indicating a high genetic variability. Genotypes varied in relation to fruit length (FL) from 44.07 to 82.40 mm. Regarding fruit diameter (FD), variation was observed between 25.23 and 38.03 mm, while pericarp thickness (PT) ranged from 2.60 to 4.37 mm.

Soluble solids (SS) content ranged from 5.87 to 9.97 °Brix, with the highest values observed for Kalani, Akamu and Kaolin. For titratable acidity (TA), Kalani genotype showed higher acidity, while Kaiki, Moke and Akamu had the lowest values. Regarding the evaluation of total carotenoid (CT) concentration, Kalani fruits also obtained the highest value (10323.08 mg beta carotene at 100 $g^{-1}$). For vitamin C (VITC), the values ranged from 183.82 to 242.65 mg of ascorbic acid in 100 $g^{-1}$, with the higher values observed in Moke, Kaolin and Kaiki with 242.65, 213.23 and 205.88 mg of ascorbic acid in 100 $g^{-1}$, respectively.

For phenolic total content (TPC), the values ranged from 677.2 to 825.5 mg GAE 100 $g^{-1}$, with the highest values observed for Akamu, Kaiki and Kaolin. These genotypes also showed high values for antioxidant activity (DPPH*, ABTS and FRAP methods). Regarding color, it is possible to affirm that Kaiki and Moke fruits are yellow, lighter and brighter, while Kalani and Kaolin fruits are red, darker and opaque than the others. The cultivar Akamu is characterized as bright orange fruits with intermediate luminosity to the yellow and red cultivars (Table 1).

**Table 1.** Physical, biochemical and sensory traits of five mini sweet pepper cultivars.

| Traits [a] | Genotypes | | | | | CV% [b] |
|---|---|---|---|---|---|---|
| | Kalani | Kaiki | Akamu | Moke | Kaolin | |
| **Physical** | | | | | | |
| FL | 82.40 a | 60.63 b | 77.73 a | 44.07 c | 48.20 bc | 9.5 |
| FD | 27.57 bc | 25.87 bc | 25.23 c | 38.03 a | 35.23 ab | 11.9 |
| PT | 3.20 b | 3.00 b | 3.23 b | 4.37 a | 2.60 b | 10.0 |
| **Biochemical** | | | | | | |
| SS | 9.97 a | 7.27 bc | 8.70 ab | 5.87 c | 8.10 abc | 12.1 |
| TA | 0.46 a | 0.29 c | 0.32 bc | 0.30 c | 0.37 b | 5.9 |
| Ratio | 21.40 c | 24.76 b | 26.84 a | 19.73 c | 21.55 c | 10.1 |
| TC | 10,323.08 a | 2374.3 b | 2758.9 b | 1707.7 b | 2066.7 b | 8.5 |
| VITC | 183.8 b | 213.2 ab | 191.2 b | 242.6 a | 205.9 ab | 8.2 |
| TPC | 677.2 c | 802.2 ab | 825.5 a | 630.5 bc | 695.5 abc | 7.4 |
| DPPH | 1389.3 bc | 1449.3 ab | 1513.5 a | 1315.3 c | 1400.5 bc | 2.8 |
| ABTS | 1489.6 bc | 1633.9 ab | 1642.0 a | 1443.1 c | 1520.8 abc | 3.5 |
| FRAP | 7597.6 b | 8059.4 a | 8097.2 a | 7127.41 c | 7692.1 ab | 2.2 |
| **Colors** | | | | | | |
| Luminosity | 33.13 c | 51.60 a | 43.37 b | 48.63 a | 30.47 c | 5.9 |
| Chroma | 31.37 b | 50.57 a | 44.33 a | 45.83. a | 22.40 c | 8.2 |
| Hue | 30.63 c | 72.33 a | 50.77 b | 71.87 a | 28.80 c | 6.9 |
| **Sensory** | | | | | | |
| Size | 8.03 a | 8.05 a | 8.03 a | 6.34 b | 7.60 a | 23.93 |
| Shape | 8.05 a | 8.22 a | 8.12 a | 6.64 b | 8.17 a | 25.27 |
| Color | 8.74 ab | 8.59 ab | 8.30 b | 8.27 b | 9.05 a | 20.33 |
| Global acceptance | 8.10 a | 8.02 a | 7.89 a | 6.70 b | 7.94 a | 25.09 |

Fruit length (FL, mm), fruit diameter (FD, mm), pericarp thickness (PT, cm), [a] total soluble solids (SST, °Brix), titratable acidity level (TA, m/m citric acid), total carotenoids (carotenoids, mg beta carotene in 100$g^{-1}$), vitamin C content (VitC, mg 100 $g^{-1}$), total phenolic content (TPC, mg GAE $g^{-1}$), total antioxidant activity by Ferric reducing antioxidant power (FRAP, mg TEAC $g^{-1}$) and 2,2-diphenyl-1-picrylhydrazyl (DPPH•, mg TEAC $g^{-1}$), and antioxidant activity by a bts method (ABTS, mg TEAC $g^{-1}$). Size, shape, color and global acceptance luminosity (L), chromaticity (C), tonality angle (h), [b] Coefficient of variation (CV). Means followed by the same letter were not significantly different at Tukey test ($p > 0.05$).

By principal component analysis (PCA), the first two components explained 86.65% of the variation (PCA 1 and PCA 2 with 48.29 and 38.37%, respectively) (Figure 2a). PCA and hierarchical Ward clustered the genotypes into three groups (Figure 2). Group I consisted of the Moke genotype that presented high values for Chroma, Luminosity, Hue, PT, FD and VITC. Group II consisted of Kaiki and Akamu genotypes that obtained the highest values for Ratio, DPPH, FRAP, ABTS and TPC, while the Group III consisted of the Kalani and Kaolin genotypes. Kalani obtained the highest values for SS, FL, TC and TA, while Kaolin obtained intermediate values.

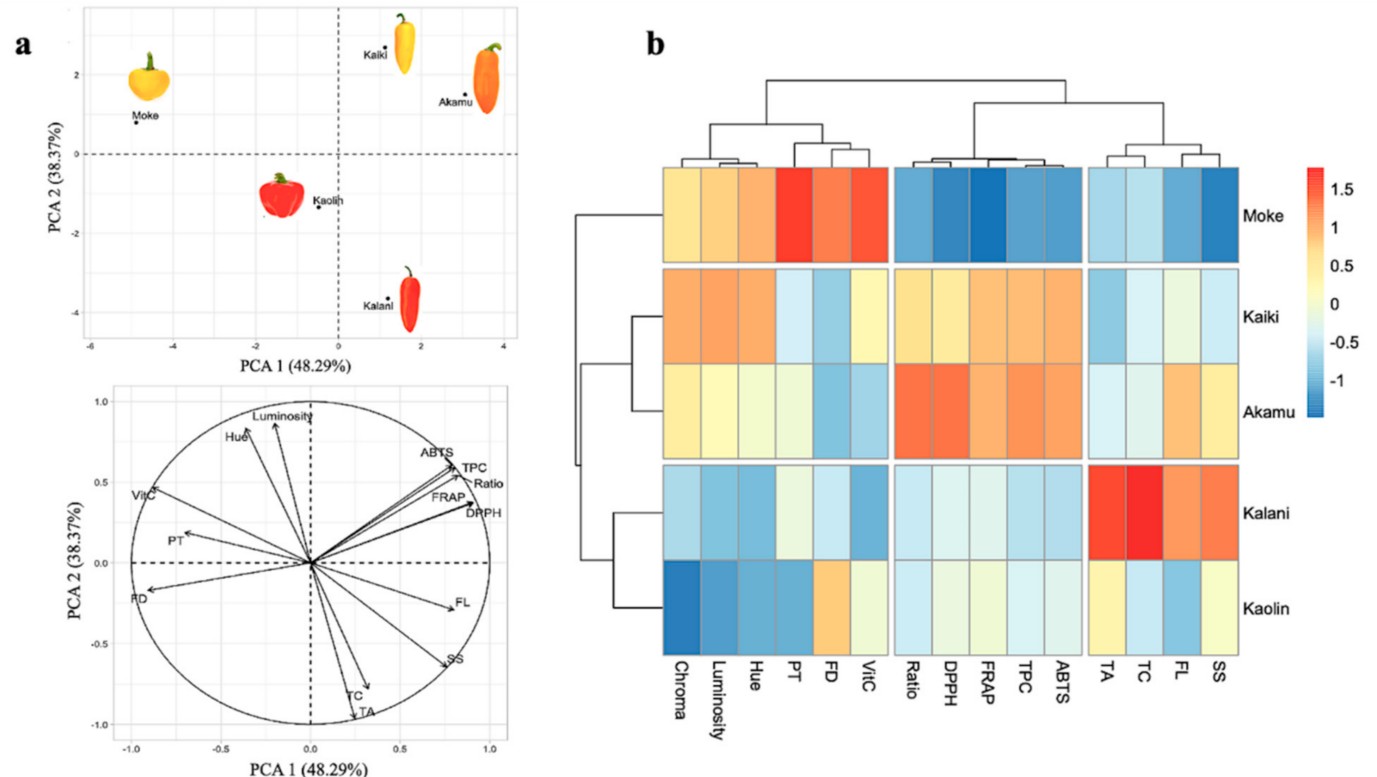

**Figure 2.** Principal component analysis (**a**) and Ward hierarchical grouping (**b**) of five mini chili genotypes for different morphological, biochemical and nutritional characteristics.Fruit length (FL, mm), fruit diameter (FD, mm), pericarp thickness (PT, cm), total soluble solids (SST, °Brix), titratable acidity level (TA, m/m citric acid), total carotenoids (carotenoids, mg beta carotene in 100 g$^{-1}$), vitamin C content (VitC, mg 100 g$^{-1}$), total phenolic content (TPC, mg GAE g$^{-1}$), total antioxidant activity by Ferric reducing antioxidant power (FRAP, mg TEAC g$^{-1}$) and 2,2-diphenyl-1-picrylhydrazyl (DPPH•, mg TEAC g$^{-1}$), and antioxidant activity by a bts method (ABTS, mg TEAC g$^{-1}$). Size, shape, color and global acceptance luminosity (L), chromaticity (C), tonality angle (h), red/green ratio (a), yellow/blue ratio (b).

The fruit firmness analyses showed differences among mini sweet peppers genotypes (Figure 3). Moke and Kaolin obtained the highest peak, indicating that these fruits require a force of approximately 1.6 and 1.34 N, respectively, to perforate the exocarp. In turn, Kaiki fruits presented the less rigid exocarp, requiring 0.97 N of force applicable until rupture. These results were concordant when the firmness of the inner layers of the fruit (mesocarp) was analyzed, in which the genotypes Moke and Kaolin obtained greater resistance to perforation. It suggests that the Kaolin and Moke fruits have the most rigid peel, and the most elastic pulp, when compared to the other genotypes.

*3.2. Sensory Evaluation*

Regarding the profile of the evaluators, 58.1% and 41.9% were female and male, respectively. Most respondents were between 26 and 35 years old (43.9%), followed by respondents aged ≤25 years (31%), between 36 and 45 years (12.3%), >56 years (7.1%) and

between 46 and 55 years (5.7%). The evaluators were questioned about the frequency with which they consumed peppers. The answers were: 49.7% for eventually consume, 16.8% for monthly consume, 18.1% biweekly, 13.5% weekly and only 1.9% answered that they consume peppers every day.

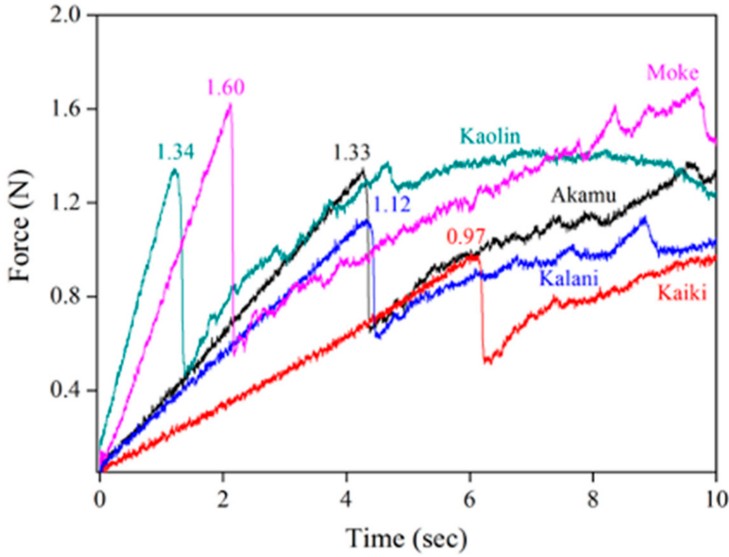

**Figure 3.** Texturogram of the five mini sweet peppers genotypes.

The lowest overall acceptance of consumers was to Moke fruits (Figure 4), while the highest acceptances for size and shape were observed for Kalani, Kaiki and Akamu. For color, Kalani and Kaolin who stood out positively. For global acceptance, higher scores were verified for Kalani, Kaiki, Kaolin and Akamu genotypes.

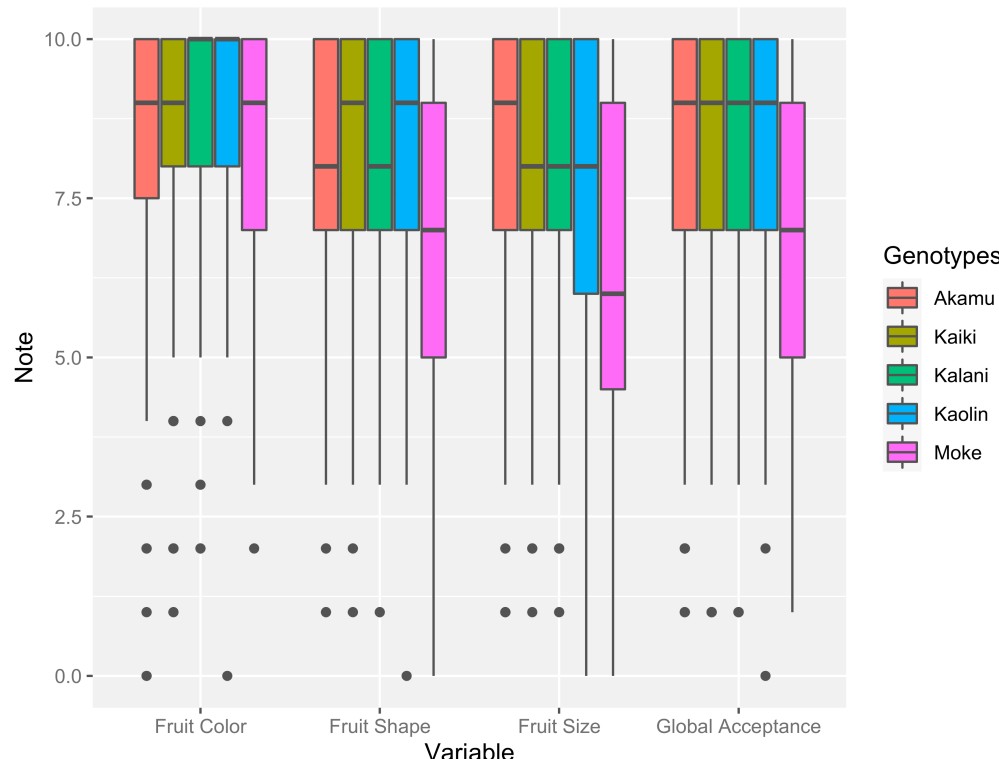

**Figure 4.** Boxplot analysis for the evaluators' scores for the sensory attributes of the five mini sweet pepper genotypes.

## 4. Discussion

The consumption of mini and baby vegetables has been growing over the years in Brazil, presenting an important appeal to consumers due to the ease of consumption and taste [8]. In addition, this type of vegetable provides a higher added value for rural producers, representing an excellent option for the diversification of production. The mini sweet peppers are new into the market with great potential, due mainly to the flavor, color, texture and shape of the fruits, being a great product to be consumed as an appetizer. Furthermore, its fruits have in their composition different substances related to improved health, such as antioxidant compounds and vitamin C, constantly related to the prevention of numerous diseases [19–21]. Therefore, this study envisioned to characterize different genotypes of mini sweet peppers regarding the physical-chemical and nutritional aspects and through sensory analysis to aware of the market potential of this vegetable.

Our study showed a wide variability among mini sweet pepper genotypes in physical, biochemical, nutritional and sensory aspects. Following [23], the commercialization of mini sweet peppers is extremely dependent on morphological variations, because the greatest acceptance comes precisely from the demand for fruit mix with different colors (yellow, orange, red and green). In this context, the use of the fruit mix can be considered an important strategy for the use of the nutritional potential of each of the genotypes evaluated.

The high SS values observed in mini sweet peppers (Kalani, Akamu and Kaolin) indicate higher natural sugar content, that is appreciated for fresh consumption as a healthy appetizer and for the industry aiming at fruit dehydration. Corrêa et al. [24], evaluating 16 sweet peppers hybrids, verified values ranging from 3.73 to 4.28 °Brix. Rinaldi et al. [25], evaluating field-grown peppers and hydroponics, obtained values from 4.90 to 7.40 °Brix. Kalani also stood out for total carotenoids, with values three to four times higher than other genotypes. Carotenoids are a group of phytochemicals responsible for the different colors of food. They have an important role for plant and animal health due to their nutritional traits and role in preventing degenerative diseases and protecting against oxidative stress [26,27]. In *C. annuum*, the main carotenoids are capsanthin, capsorubin, β-carotene, zeaxanthin, violaxanthin, lutein and antheraxanthin, which vary in concentration in different stages of fruit maturation [28,29].

The high vitamin C values found in the mini sweet peppers indicates their great nutritional potential, and it is an important trait for exploration in marketing. Mennella et al. [30], evaluating 15 genotypes of sweet peppers, found values ranging from 105.11 to 275 mg of ascorbic acid at 100 $g^{-1}$ with an average of 167.47 mg of ascorbic acid at 100 $g^{-1}$. Nankar et al. [31], evaluating 180 accessions of *C. annuum*, found values ranging from 4.77 to 273.47 mg of ascorbic acid at 100 $g^{-1}$ with an average of 115.59 mg of ascorbic acid at 100 $g^{-1}$. In the present study, the average vitamin C was 207.35 mg ascorbic acid at 100 $g^{-1}$, 3.5 times the recommended daily value for adults who consume 2000 calories [32].

Sweet pepper fruits are also rich in phenolic compounds, which are beneficial to health due to their ability to eliminate free radicals in biological systems in vitro and in vivo [26,33]. This fact was corroborated by the high correlation between CPT and methods that determine antioxidant capacity in vitro (DPPH, FRAP e ABTS), where Kaiki and Akamu fruits presented the may values. Antioxidants are known as cellular protectors for the fight of free radicals in the body, not allowing the oxidation process. Series of free radical-initiated reactions cause membrane data, disrupting metabolic pathways, thereby increasing DNA mutations and changes in platelets and other functions [34].

The greater firmness of the fruits may be related to the shape of the most found fruit, requiring greater strength for the disruption of the exocarp and mesocarp. However, this shape of the fruit presented lower acceptance among the interviewees. Vision is an important parameter in sensory analysis, because it is the first impressions of the products. Despite the small size of these vegetables, the market for mini sweet peppers has gradually increased and increasingly conquers the consumer. At the same time, the development of cultivars that present special characteristics is required by the industry, not only in relation to fruit morphology, such as attractive nutritional size and nutritional format,

but also by selecting cultivars that unite all the benefits that the species can provide. The appearance of a product is one of the most important factors evaluated by the consumer at the time of purchase and; therefore, a mix of fruits of different colors and elongated shape of mini sweet peppers can be considered an important strategy for commercialization, associating with the nutritional quality of this group of vegetables. Summing up, our results provides excellent information for research and industry about the characteristics of the mini sweet peppers. These data are important to provide pathways and assist in the creation of strategies for both improvement and trade, to the development of more nutritious and attractive vegetables. Therefore, it also can be applied to several other classes of mini vegetables.

## 5. Conclusions

A wide variability was observed among genotypes for physical-chemical and nutritional traits. Based on sensory analysis, Kalani, Kaiki, Kaolin and Akamu obtained greater global acceptance. The data that have been presented here are just the beginning of a characterization about the mini sweet pepper genotypes. These mini vegetables have greater potential to be explored and accordingly several other analyses are necessary to better understand and get information about their characteristics and qualities and thus design new perspectives and strategies for improving and increasing the consumer market. The genotypes can be considered an important marketing strategy of mini sweet peppers trade, associating different shapes, colors and nutritional quality. Whereas several public policies and innovations in the agricultural sector have been carried out in order to encourage the consumption of vegetables, this information can contribute to the development programs of new cultivars, with focused efforts on mini vegetables that are not only attractive by shape and color, but also associate with the excellent nutritional characteristics.

**Author Contributions:** L.V.C. and L.S.A.G. conceived the idea and designed the study. M.B.C.R., A.F.N., K.S.B., L.M.R. and A.M.M. performed the experiment; D.M.Z. analyzed the data and built the figures. R.M.G. and L.V.C. wrote and translated the manuscript, while L.S.A.G. carried out text final revision. All authors discussed the results, conceived, and approved the final manuscript. All authors have read and agreed to the published version of the manuscript.

**Funding:** This study was financed in part by the Coordenação de Aperfeiçoamento de Pessoal de Nível Superior-Brasil (CAPES)-Finance Code 001.

**Institutional Review Board Statement:** The study was conducted according to the guidelines of the Declaration of Helsinki, and approved by the Ethics Committee of Universidade Estadual de Londrina (protocol code 3.028.485 and aproved on november of 2018).

**Informed Consent Statement:** Informed consent was obtained from all subjects involved in the study.

**Conflicts of Interest:** The authors declare that they have no conflict of interest.

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
