# Peer review of "Post-Harvest Quality and Sensory Evaluation of Mini Sweet Peppers"

_horticulturae, doi:10.3390/horticulturae7090287_

Round 1
Reviewer 1 Report
The manuscript describes the chemical characterization of different genotypes of sweet peppers. Data reported are not highly informative, as they could have been much more detailed. By a nutritional point of view, the mineral composition, including microelements, would worth attention, as well as the detailed carotenoids abundance (with quantitation of each carotenoid). Also a more detailed investigation about antioxidants and other bioactive molecules (flavonoids, anthocyans, etc) would greatly improve the manuscript, making it much more informative.
Author Response
Please see the attachment.
Best regards,

Reviewer 2 Report
It would be interesting for the authors to specify after harvesting, what were the storage conditions of the mini-peppers(temperature, humidity, aerobic conditions, presence of visible / UV radiation, ...) before proceeding with their physical
characterization and Biochemistry It must be taken into account that vitamin C,phenolic compounds and the antioxidant capacity
of mini-peppers are very susceptible to undesirable oxidation processes that can significantly modify the nutritional and sensory
value (lines 80-89 ).
When considering marketing strategies for mini-peppers, they should first establish what would be the most suitable packaging
systems to avoid a decrease in nutritional and sensory value due to unwanted chemical oxidation reactions that will occur during
the period of storage at places of sale and marketing.
Author Response

(The authors gave the same response as above.)

Reviewer 3 Report
This manuscript provides a report on post-harvest quality and sensory acceptance of mini sweet peppers. The manuscript is of value for publication but it requires some additions.
In order for the title to be properly scientific, I suggest that the Latin name of the pepper should be also included. I also suggest using „sensory evaluation” instead of „sensory acceptance” not only in the title but throughout the manuscript.
In the literature review section, a broader review of mini vegetables is suggested, this is an interesting topic and is widespread in many more vegetable species.
In the Materials and Methods part as there was no temporal or spatial repetition of the experiment, I consider it more appropriate to describe the experimental conditions more precisely, like the laboratory preparation of samples for measurements.
The presentation of the results of sensory evaluations is not scientifically well-founded, in the manuscript, I suggest rethinking the presentation and evaluation of these results.
There is no critical discussion of the results. To be novel, a paper must state how the research has contributed to theory. However, you have the basis of some good information to provide to industry as recommendations, try to emphasize this much better.
The conclusion part is too short and does not really contain new scientific findings but rather only findings about the plant material used in the experiment.
I would recommend a resubmission of the paper. To do so, they have to:
- Present that this is a study that’s going to a practical use
- Present all the growing conditions (temperature, irrigation, fertilization etc.)
- Emphasise the accuracy of harvesting stage on fruit quality parameters
- Presenting all accurate sample preparation during laboratory measurements (e.g. from which part of the pepper the parameter was measured, how many pieces of fruit in how many repetitions etc,)
- Rewriting the presentation and evaluation of sensory results
- In Conculison part try to emphasize much better your results don’t just include a general statement.
- In References indicate the year and journal where it was published, there is a lot of inaccuracy in the list
Author Response

(The authors gave the same response as above.)

Round 2
Reviewer 1 Report
Dear authors,
the pandemic has been very hard for all of us; however, this cannot justify the lack of analyses and results.
I well understand your point of view, but we have to keep in mind that a scientific paper has to be focused on scientific results and cannot be limited to the willingness of sensitizing/educate people to vegetable consumption and healthy diet.
My opinion is still that the manuscript should be improved.
Reviewer 3 Report
The manuscript is a resubmission and appropriately improved version.
I accept the responses.
Author Response
Dear Reviewer,
We kindly thank you for your time and consideration.
Best regards,